# Life Cycle Assessment of the Domestic Micro Heat and Power Generation Proton Exchange Membrane Fuel Cell in Comparison with the Gas Condensing Boiler Plus Electricity from the Grid

Lyubov Slotyuk [1], Florian Part [2], Moritz-Caspar Schlegel [1] and Floris Akkerman [1,*]

1 Fachbereich Ökodesign und Energieverbrauchskennzeichnung, Bundesanstalt für Materialforschung und -Prüfung (BAM), Unter den Eichen 82, 12203 Berlin, Germany; lyubov.slotyuk@bam.de (L.S.); moritz-caspar.schlegel@bam.de (M.-C.S.)

2 Department of Water-Atmosphere-Environment, Institute of Waste Management and Circularity, University of Natural Resources and Life Sciences, Muthgasse 107, 1190 Vienna, Austria; florian.part@boku.ac.at

* Correspondence: floris.akkerman@bam.de

**Abstract:** The energy demand of private households contributes globally to 36.5% of the total $CO_2$ emissions. To analyze the emissions reduction potential, we conducted a comparative life cycle assessment of a proton exchange membrane fuel cell in a residential application and a conventional system with a stand-alone gas condensing boiler and electricity from a grid mix. The period under review was referred to as the service life of the PEMFC and is assumed to be 10 years (83,038 h of PEMFC). The applicability of this in a single-family house built between 1991 and 2000 under German climatic conditions was investigated. The functional unit is set to the thermal energy demand of 16,244 kWh/a and electricity demand of 4919 kWh/a of a single-family house. The impact assessment method "CML 2001–August 2016" was used in this investigation. The manufacturing phase of the proton exchange membrane fuel cell showed disadvantages, whereby the use phase had significant advantages in most of the environmental impact categories as compared to the conventional energy supply system. Considering the whole life cycle, the advantages from the use phase could outperform the disadvantages from the manufacturing phase in most of the impact categories, except for ADP elements and TETP.

**Keywords:** sustainability; comparative life cycle assessment; proton exchange membrane fuel cell; gas condensing boiler; micro heat and power generation; residential application; single-family house energy supply system

## 1. Introduction

Electricity and heat generation are the largest sources of global greenhouse gas emissions worldwide, accounting for about 55% [1,2]. The residential sector accounts for 39% of global final energy consumption in buildings and industry [3]. The majority (82%) of the total primary energy demand in the EU residential sector is consumed for space heating and hot water supply [4,5]. In line with the goals of the European Union of "improving energy efficiency" by 27% in 2030 and "reducing greenhouse gas emissions" by 40% in 2030 compared to 1991, the minimum efficiency requirements within the Ecodesign legislative framework and the EU Energy Label [6–8] have proven to be effective. Another effective tool of eco-design, which has been gaining importance in recent years, is life cycle assessment (LCA), which allows for the quantitative assessment of the life cycle environmental performance of products and can be used by manufacturers and developers to design more sustainable and energy-efficient products, as well as by the government to support decision-making for legislation [9].

In addition to energy efficiency, increasing the share of low-emission energy technologies is another way to optimize the environmental impact of the energy sector [10].

However, the market for heating appliances continues to be dominated by fossil fuel-based appliances and less efficient conventional electric heating technologies, which account for almost 80% of new sales [11]. Sales of heat pumps and renewable heating devices such as solar hot water systems have nevertheless increased, accounting for more than 10% of total sales in 2019. To be in line with the International Energy Agency's (IEA) sustainable development scenario (SDS), the share of low-emission heating technologies, such as heat pumps, district heating, and renewable and hydrogen-based heat, must increase to 50% of sales by 2030 [11,12]. Among various clean energy supply technologies, fuel cells (FC) seem to be a promising alternative to conventional energy supply systems because of relatively high energy efficiency and low emissions [13–16]. Several support packages were developed in Europe, generally, and especially in Germany to encourage the residential use of fuel cell technology. Consequently, more than 1000 systems have been built in 10 countries, with the largest number of installed units in Germany [6].

Bachmann et al., 2019 conducted the LCA of residential fuel cell micro heat and power cogeneration (FC-μCHPs) and compared it with a stand-alone gas condensing boiler (GCB) and a heat pump (HP) [17]. For the assumed full loading hours (FLHs), relevant environmental impacts, including global warming potential, were generally smaller for the FC-μCHPs than for the HP and the stand-alone GCB [17]. Notter et al., 2015 conducted a comparative cradle-to-grave LCA of two types of Proton-exchange membrane fuel cell (PEMFC) applications: PEMFC μCHP (with carbon black (CB) and multiwalled carbon nanotubes (MWCNTs)) and Stirling engine. Their findings are that platinum is the key material in HT-PEMFCs, whereby the benefits from platinum savings outperform the burdens from MWCNT production. Furthermore, they found out that both μCHP plants (PEMFC and Stirling engine) have comparable environmental impacts. However, the PEFMC produces more electricity and less heat as compared to the Stirling engine. System expansion in the way that both plants produce an equal amount of electricity and heat resulted in an advantage of 20%. Points ReCiPe for the PEMFC [18]. Other LCA studies show that FCs are only more environmentally friendly and efficient in comparison to conventional systems if the operation or use phase is considered [19,20]. The main role here is fuel sourcing, where hydrogen production causes environmental impacts. Concerning the manufacturing phase, the same applies to electrode materials, where, in many cases, either a noble metal such as platinum or other precious metals or costly materials are used [19,20]. Stack optimization and research for eco-friendly materials are important steps to achieve this goal [21]. Furthermore, the results of Lotrič et al., 2021 and Mori et al., 2021 show that the environmental impacts of the manufacturing phase could be substantially reduced by recycling the used device components and materials [22,23]. The authors pointed out the importance of critical materials—in this case, the platinum-group metals (PGMs) by comparing the end-of-life phase with and without the recycling of PGMs. The comparative LCA results showed that the environmental impacts increase in the case of both systems (alkaline water electrolyzer (AWE) and proton-exchange-membrane water electrolyzer (PEMWE)) because of the larger quantity of PGMs compared to the scenario without the recycling of PGMs [22]. Also, Riemer et al., 2023 [24] conducted a comparative LCA study on commercial state-of-the-art PEMFC with high and low platinum content and a novel technology called anion exchange membrane fuel cell (AEMFC) regarding PGM loading, cell performance, and lifetime. The authors' findings were that the PEMFC outperforms the AEMFC from an environmental point of view despite the lower PGM loading of the AEMFC. Increasing the performance and lifetime of AEMFC resulted in a lower or comparable environmental impact than a PEMFC in 17 out of 27 of impact categories, including climate change. The main contributors to the environmental impact of both systems are platinum in the electrodes, chromium steel in the bipolar plates, and polytetrafluoroethylene (PTFE) in the gas diffusion layer [24]. Parise et al., 2005 conclude that there is still a long way to go for the widespread use of fuel cells, which depends on cost, efficiency, and life expectancy [15,21,25]. According to the reviewed literature, the durability of the PEMFC for stationary application may vary between ca. 40,000 and

80,000 h [3,26]. As stated in the literature cited by Dhimish et al., 2021, there could be a linkage between the degradation rate due to low voltage produced by the fuel cell and the membrane temperature [25]. As stated by Yan et al., 2023, the large-scale commercialization of PEMFCs requires higher power and current densities; however, at high operating current densities, the massive accumulation of liquid water will lead to flooding and impede the gas diffusion, resulting in rapid degradation of cell performance. Accordingly, improving the water management ability is imperative for pursuing better cell output performance [27]. Furthermore, the operational strategy (electricity- vs. thermal-led strategy) could impact the performance, economic aspects, and environmental profile of the PEMFC-based CHP system [15]. From the environmental point of view, the thermal-led strategy, which was investigated in their current work, proved to be more advantageous compared to other alternatives [15].

In summary, proton exchange membrane fuel cell (PEMFC) systems are considered promising solutions for producing clean energy [19,28–30]. However, only a few comparative LCA studies have been conducted that show significant reductions in $CO_2$ emissions compared to conventional and other alternative energy supply technologies [3,18,31–33]. Therefore, our LCA study focused on the global warming potential as well as on the other environmental aspects regarding critical raw materials and recyclability. In contrast to the mentioned LCA studies [3,18,31–33], we have analyzed a semi-self-sufficient system (PEMFC with the integrated lead battery as an intermediate storage) that is suitable under certain circumstances to cover the entire energy demand of a single-family house. This comparative LCA was conducted for the case of German households to identify environmental hotspots and gain information for decision-makers such as developers and manufacturers of PEMFC systems, consumers, or authorities.

## 2. Materials and Methods

The LCA methodology, according to the ISO standards 14040 and 14044 [34,35], is used to analyze the above-defined systems throughout their entire life cycle. The FC-Hy guide reports [35–37], which provide an overview of the LCA performing on FC technologies, as well as the Sphera LCA manual on Database and Modelling Principles [38] and all guidelines from the ILCD handbooks [39], were considered and used in this analysis.

In the current work, a system consisting of HTPEMFC (100–200 °C) with a phosphoric acid (PA)-doped polybenzimidazole (PBI) membrane and the lead battery as an intermediate storage in combination with a GCB was compared with a conventional energy supply system based on the stand-alone GCB and electricity from grid mix. The PEMFC is one of the most highly developed fuel cell technologies [28,40,41]. To date, two types of proton conductive membranes have been used in the PEMFC systems for CHP applications: the perfluorosulfonic acid (PFSA) membrane (e.g., DuPont's Nafion® membrane) and phosphoric acid (PA)-doped polybenzimidazole (PBI) membrane [42]. As compared to PFSA membrane-based low-temperature (LT)-PEMFC (systems, the salient features of the HT-PEMFC systems using PA-doped PBI membranes include simpler water management, more effective waste heat utilization, and superior CO tolerance (<1%) [16,42–44]. This makes the HT-PEMFC more advantageous and suitable for the combined heat and power (CHP) generation [16,42,43,45]. So far, lithium-ion (Li-ion) and lead-acid are two of the most used batteries in stationary applications as intermediate storage. We decided to use a lead-acid battery in our model because of its popularity for such applications due to its low cost and simple charging properties [46]. Furthermore, this assumption represents a worst-case assumptions approach of LCA methodology [38] as the lead-acid battery results in a worse environmental performance as compared to the Li-ion battery [46]. The case study was conducted for Germany, and the German grid mix electricity was used for this attributional life cycle assessment (LCA) using the software Sphera LCA for experts (former GaBi—Product Sustainability Software (version 10.6.2.9)) including the Sphera LCA database (content version 2022.2) [47].

For this model, the system boundaries were chosen to consider both heat and electricity generation and utilization for a single-family house. The functional unit is defined as the total energy demand for electricity supply, space heating, and hot water for a single-family house of 124 sqm. in Germany over the period of 10 years. To analyze the environmental impact and the ability of the system to cover the entire energy demand under different circumstances, we have conducted sensitivity analyses on the different building types. The basis scenario was assumed for the building type built between 1991 and 2000. In addition, sensitivity scenarios for the old building (built between 1949 and 1978) and the new building (built after 2009) were conducted. To address the durability and degradation rate issues of the PEMFC technology, as stated in the reviewed literature [3,26], the second sensitivity analysis was performed to analyze its effect on the environmental impact. All the energy and material flows of the investigated system are recorded as comprehensively as possible and related to the functional unit. The life cycle inventory (LCI) (from Sphera LCA for Experts (GaBi) Databases content version 2022.2) in this study covers the entire life cycle, i.e., extraction of raw materials, provision of energy sources, production of intermediate products and products, use, and end of life [47].

The system definition, as schematically illustrated in Figure S1 of the Supplementary Information, was based on data from scientific literature and publicly available data sheets from fuel cell manufacturers [3,22,26,48,49] as well as generic background data from GaBi datasets [47]. The life cycle inventory (LCI) of the HT-PEMFC system is based on the data from Lotrič et al., 2021 and Mori et al., 2021 [22,23] and is presented in Table S1 of the Supplementary Information. The LCI data for the production presented in [22,23] were for the HT-PEMFC with the lead battery as an intermediate storage and 5 kW electrical power. Therefore, an assumption of linear scaling with a scaling factor of 0.15 to 750 W electrical power was undertaken [22,23]. The LCI for the SMA is based on primary data. The commercially available SMA Model Sunny Tripower inverter was procured and manually dismantled for component analysis. The LCI of the Inverter is presented in Tables S2–S4 of the Supplementary Information. To calculate the use phase, the technical performance of the commercially available HT-PEMFC device Vitovalor PT2 was used (please see Table S5 of the Supplementary Information and Table S7 with the use phase parameters used in the LCA model) as well as the assumptions for the durability and degradation of the PEMFC based on the reviewed literature (Table S6 of the Supplementary Information) [3,26,48,49]. For the basis scenario (building type built between 1991–2000) and an old building scenario (built between 1949 and 1978), the HT-PEMFC is operated in the full loading hours (45.5 h operation and 2.5 h regeneration time), and for the new building scenario (built after 2009)— in the part loading hours (15 h/day) and therefore is supported by additional electricity from a German grid mix. The heat losses and the changes in the electrical efficiency due to the part loading hours operation were neglected in this assessment. Assumptions for the use of phase basis scenarios and sensitivity scenarios can be found in Tables S8 and S9 of the Supplementary Information. The assumptions for the second sensitivity scenario on the degradation rate and durability of the PEMFC are presented in Table S10 of the Supplementary Information. A peak-load GCB with a capacity of 20 kW and an average weight of 133 kg is used to support the fuel cell-based energy supply system. The inventory is taken from the Sphera LCA for Experts (GaBi) database [47]. The efficiency of the boiler reaches 95% when considering the higher heating value of natural gas. For simplicity, the energy supply system is assumed to be maintenance-free. The flow charts of the parameterized Sphera LCA for Experts (GaBi) model for manufacturing and use phases are presented in Figures S2–S8 of the Supplementary Information.

The end-of-life model for the PEM fuel cell is based on market data [22,23]. Recyclable materials from the investigated energy supply system were considered, and thus, environmental burdens resulting from recycling processes are attributed to consumers of secondary materials. The flow charts of the parameterized Sphera LCA for Experts (GaBi) model for the end-of-life phase are presented in Figures S9–S14 of the Supplementary Information.

The midpoint approach "CML 2001–August 2016" methodology developed by the University of Leiden was used for the life cycle impact assessment (LCIA) [50]. The environmental impacts were analyzed for the following impact categories: abiotic depletion potential (ADP elements and ADP fossil), acidification potential (AP), eutrophication potential (EP), freshwater aquatic ecotoxicity potential (FAETP), global warming potential (GWP), human toxicity potential (HTP), marine aquatic ecotoxicity potential (MAETP), ozone layer depletion potential (ODP), photo-chemical oxidation potential (POCP), and terrestrial ecotoxicity potential (TETP), as well as primary energy demand (PED). These are the most used in the EU guidelines and are also very important in evaluating FC technologies throughout all the life cycle phases [22].

Equation (1) was used for normalization of the LCA results to compare the two different energy supply systems. For the schematic illustrations, the GCB and electricity from grid mix system is presented as a reference system and is set as 100%. Whenever the results are negative, this is due to the electricity credits. The absolute results for all impact categories are presented in the Supplementary Information.

$$\text{Normalized LCA results} = \frac{\text{impact of PEMFC system} - \text{impact of GCB plus German grid mix electricity}}{\text{impact of PEMFC system}} \times 100\% \quad (1)$$

## 3. Results

### 3.1. PEMFC Compared to a Conventional Energy Supply System

Figure 1 presents a general overview of the normalized LCA results of the conventional system (i.e., stand-alone GCB and German grid mix electricity) and innovative system (PEMFC with GCB). The absolute values of these results can also be found in the Table S11 of the Supplementary Information. The PEMFC-based system shows advantages as compared to the conventional GCB with grid mix system in almost all of the considered impact categories, except for ADP elements and TETP. For the GWP, the reduction amounts to 76% (65,607 and 15,510 kg $CO_2$-eq. in absolute values). Concerning the ADP elements, the PEMFC-based system showed significant disadvantages—117% higher contribution. This is due to the PGMs contained in the stack and the balance of plant of the PEMFC, as well as lead-containing in the battery storage.

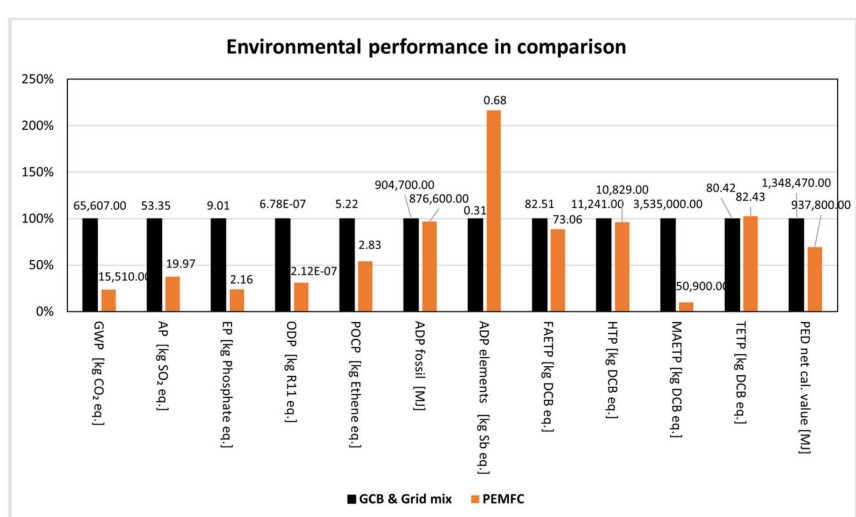

**Figure 1.** Normalized life cycle assessment results of the energy supply system for a single-family house in Germany: gas condensing boiler (GCB) and electricity from grid mix compared with the proton exchange membrane fuel cell (PEMFC).

Figure 2 presents a detailed overview of the LCA results for manufacturing, use, and end-of-life phases of (A) the innovative system (PEMFC with GCB) and (B) the conventional system (i.e., stand-alone GCB and German grid mix electricity). The absolute values for

these results are presented in the Table S11 of the Supplementary Information. In the case of the PEMFC with GCB system for many of the considered impact categories (except for GWP, ADP fossil, TETP, and PED), the manufacturing phase shows the dominating contribution to the total results. Whereby, for the GWP, the absolute value accounts for 2920 kg $CO_2$-eq. However, in the case of the stand-alone GCB and Grid mix system, the use phase is a main contributor for all impact categories, except for ADP elements.

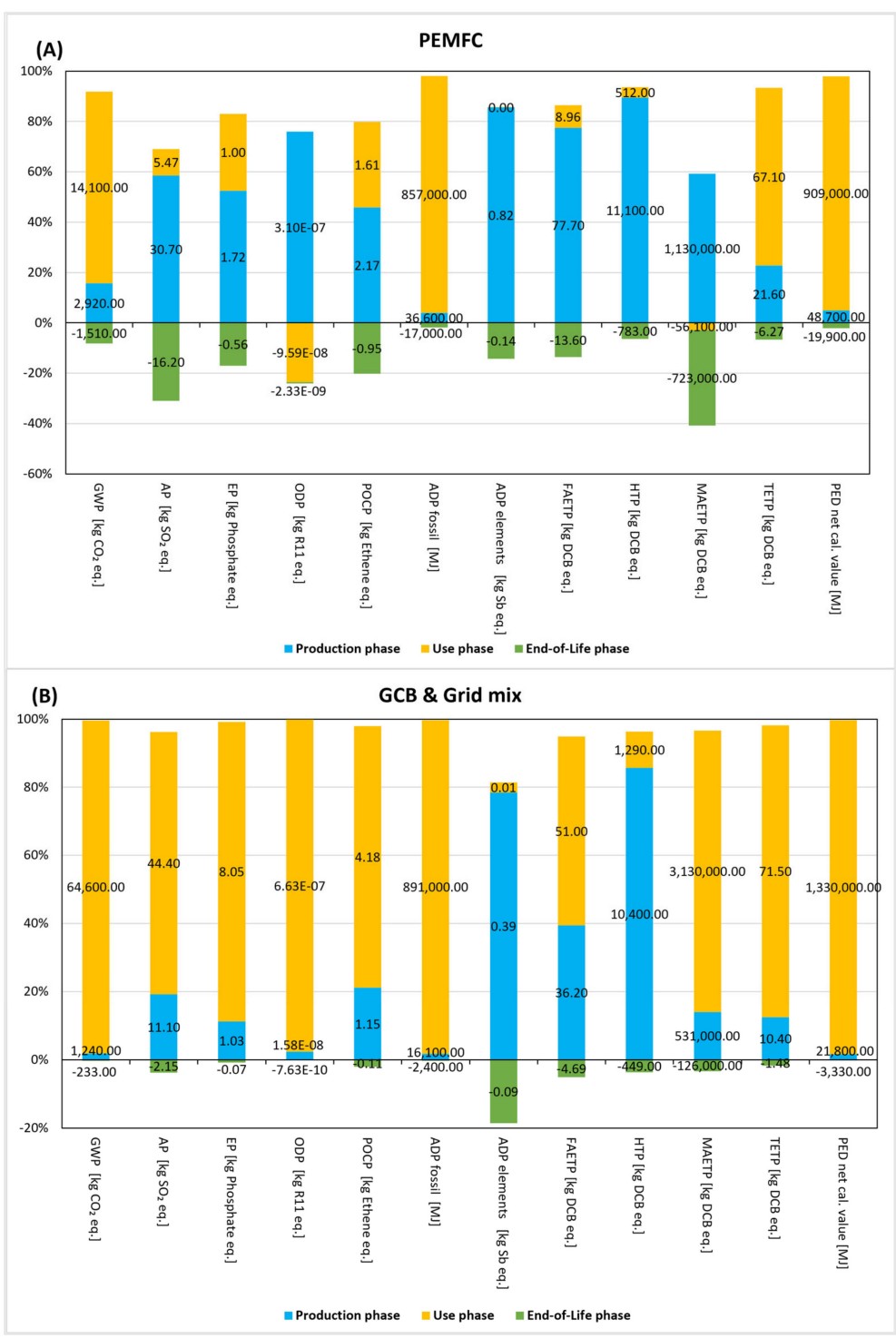

**Figure 2.** Detailed overview of life cycle assessment results of the energy supply system for a single-family house at German climatic conditions: Proton exchange membrane fuel cell (PEMFC) based system (**A**) compared with a gas condensing boiler (GCB) and electricity from grid mix (**B**).

### 3.2. Environmental Profile of the PEMFC during the Manufacturing Phase

The manufacturing phase was analyzed in more detail to identify the environmental hotspots of the whole PEMFC energy supply system, which is shown in Figure 3. The absolute values for these results are presented in the Table S12 of the Supplementary Information. The PEMFC dominates among all other energy supply system components in almost all impact categories (between 37% and 83%) except for the HTP and TETP categories (9 and 18%, respectively). Whereby the absolute value for the GWP accounts for 1500 kg $CO_2$-eq. This is led back to the use of precious metals in the stack and supporting components of the balance of the plant (43.6% contribution to the total GWP of the PEMFC). The underfloor heating system made of copper is the second dominant contributor to the total results in almost all impact categories (between 16% and 40%) except for HTP, ODP, and PED categories. The absolute value for the GWP amounts to 499 kg $CO_2$-eq. The HTP category is dominated by the manufacturing of the water reservoir (78.7%). This is due to the chromium and nickel exposure as well as the carcinogenicity during the mining and ram material extraction of stainless steel [51].

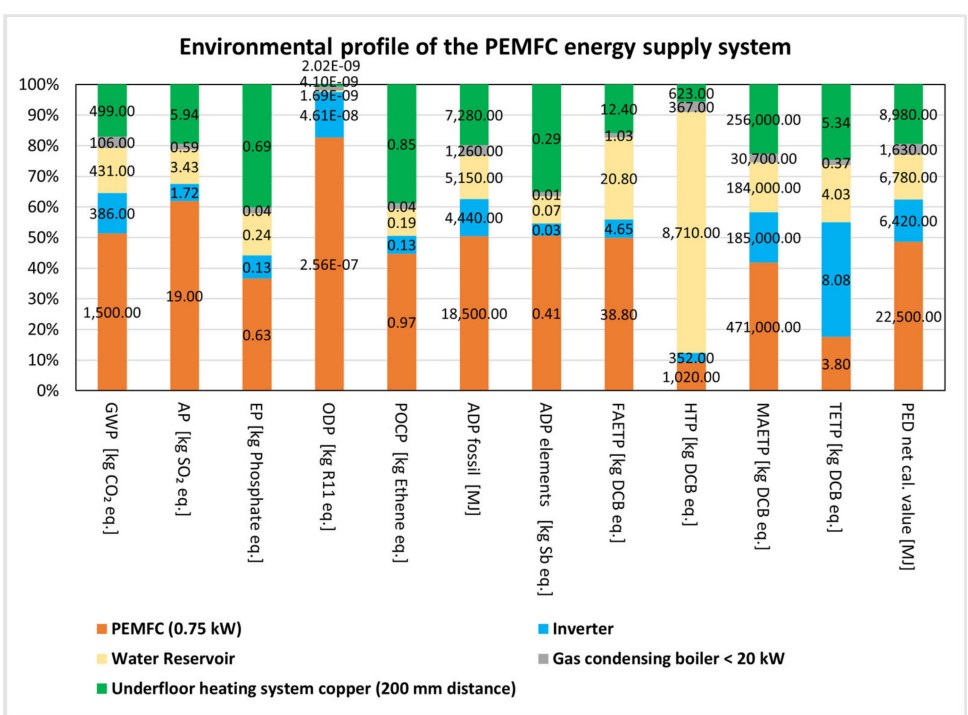

**Figure 3.** Detailed overview of the life cycle assessment results of the manufacturing phase of the PEM fuel cell-based system.

### 3.3. Comparative Life Cycle Assessment Results: Use Phase

Figure 4 shows the overall normalized results, and Figure 5 shows a detailed overview of the results for both energy supply systems. The absolute values for these results are also presented in the Table S13 of the Supplementary Information). The results for the PEMFC system include the credits for the surplus electricity, which is produced by the PEMFC and fed into the grid mix (see Table S9 of the Supplementary Information). As can be seen in Figure 5, the innovative PEMFC system shows advantages from an environmental point of view in all investigated impact categories. The emissions reduction for GWP accounts for 78% (64,600 kg $CO_2$-eq. versus 14,150 kg $CO_2$-eq. for the conventional and innovative systems, respectively).

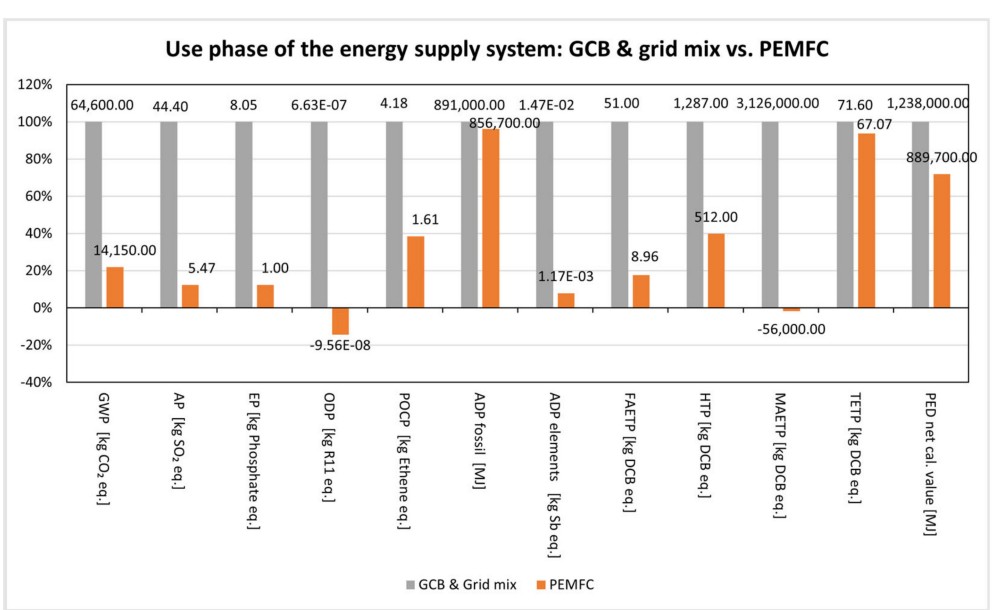

**Figure 4.** Normalized results of comparative life cycle assessment results of use phase of the energy supply system for a single-family house at German climatic conditions: Proton exchange membrane fuel cell (PEMFC) based system versus gas condensing boiler (GCB) and electricity from grid mix. The GCB and electricity from the grid mix system are presented as a reference system and are set as 100%.

Taking a look at a detailed overview of the results for the use phase in Figure 5, one can see that in the case of the PEMFC-based energy supply system, the natural gas for fuel cells contributes most to all impact categories (between 62% and 80%) except for the GWP (27%) and the ODP (26%), whereas the contribution from GCB (73% and 74%, accordingly) is dominated. The absolute values for the GWP amount for 4940 kg $CO_2$-eq. and 13,400 kg $CO_2$-eq. from the natural gas for the fuel cell and from the GCB, including natural gas, respectively. It should be noted that in Figure 5, only the environmental burdens, excluding credits for the surplus electricity for the PEMFC system, are presented. Whereas in the case of the PEMFC system, 828 kWh/a surplus electricity is fed into the grid mix, which can be credited and improve the environmental profile of the PEMFC system. The absolute credit values are presented in Table 1 below. In the case of the GCB and electricity from grid mix energy supply system (conventional system), the contribution from electricity grid mix dominates in all impact categories except for GWP, ADP fossil, TETP, and PED, where the contribution from the gas condensing boiler is dominated.

**Table 1.** Electricity credits for the surplus electricity of 828 kWh/a produced by the PEMFC and fed into the grid mix.

| Impact Categories | Electricity Credits |
|---|---|
| GWP [kg $CO_2$-eq.] | $-4.19 \times 10^3$ |
| AP [kg $SO_2$-eq.] | $-5.39$ |
| EP [kg Phosphate eq.] | $-1.01$ |
| ODP [kg R11 eq.] | $-1.07 \times 10^{-7}$ |
| POCP [kg Ethene eq.] | $-3.74 \times 10^{-1}$ |
| ADP fossil [MJ] | $-4.13 \times 10^4$ |
| ADP elements [kg Sb eq.] | $-2.07 \times 10^{-3}$ |
| FAETP [kg DCB eq.] | $-6.46$ |
| HTP [kg DCB eq.] | $-1.41 \times 10^2$ |

**Table 1.** *Cont.*

| Impact Categories | Electricity Credits |
| --- | --- |
| MAETP [kg DCB eq.] | $-4.60 \times 10^5$ |
| TETP [kg DCB eq.] | $-4.73$ |
| PED net cal. value [MJ] | $-9.53 \times 10^4$ |

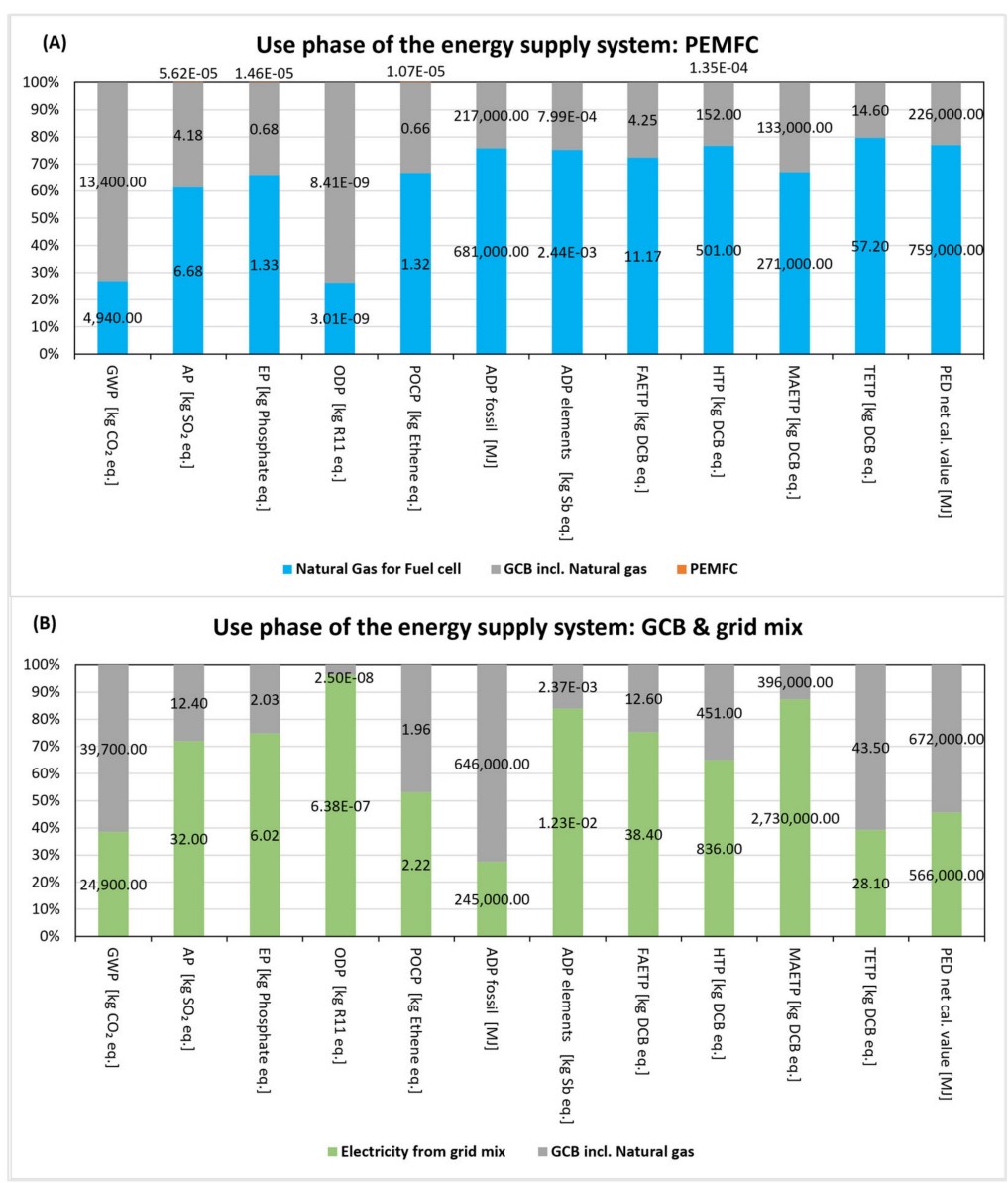

**Figure 5.** Detailed overview of life cycle assessment results of the use phase of the energy supply system for a single-family house at German climatic conditions: Proton exchange membrane fuel cell (PEMFC) based system (**A**) compared with a gas condensing boiler (GCB) and electricity from grid mix (**B**).

### 3.4. Sensitivity Analyses

Two types of sensitivity analyses were performed in this study: (1) for the entire life cycle of the PEMFC energy supply system operated in the different building types (building between 1991 and 2000 as basis scenario, new building from 2009 and old building between 1949 and 1978); (2) for the entire life cycle of the PEMFC energy supply system with different durability of the PEMFC (10 years or 83,038 operating hours as a basis scenario; and 5 years or 41,519 operating hours as an alternative scenario). The assumptions for the performed

sensitivity analyses are presented in Tables S9 and S10 of the Supplementary Information. The results of these sensitivity analyses are shown in Figures 6 and 7 below. The absolute values for the sensitivity analysis results can be found in the Tables S14 and S15 of the Supplementary Information.

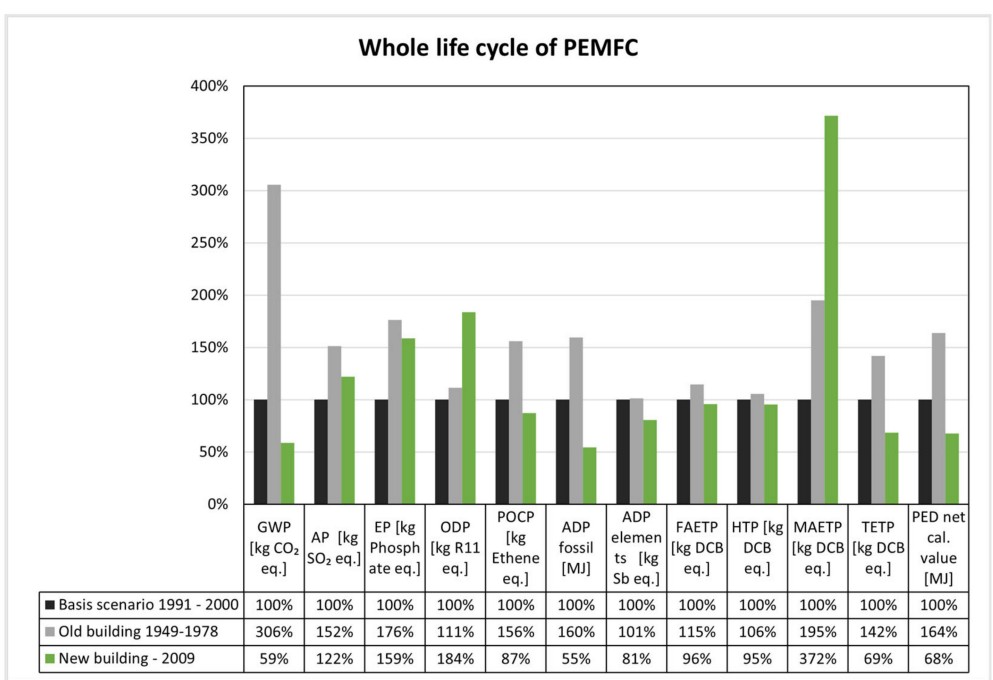

**Figure 6.** Sensitivity scenario LCA of the PEM fuel cell energy supply system: comparison of basis scenario building type from 1991–2000 with the old building type from 1949–1978 and new building type from 2009. The GCB and electricity from the grid mix system are presented as a reference system and are set as 100%.

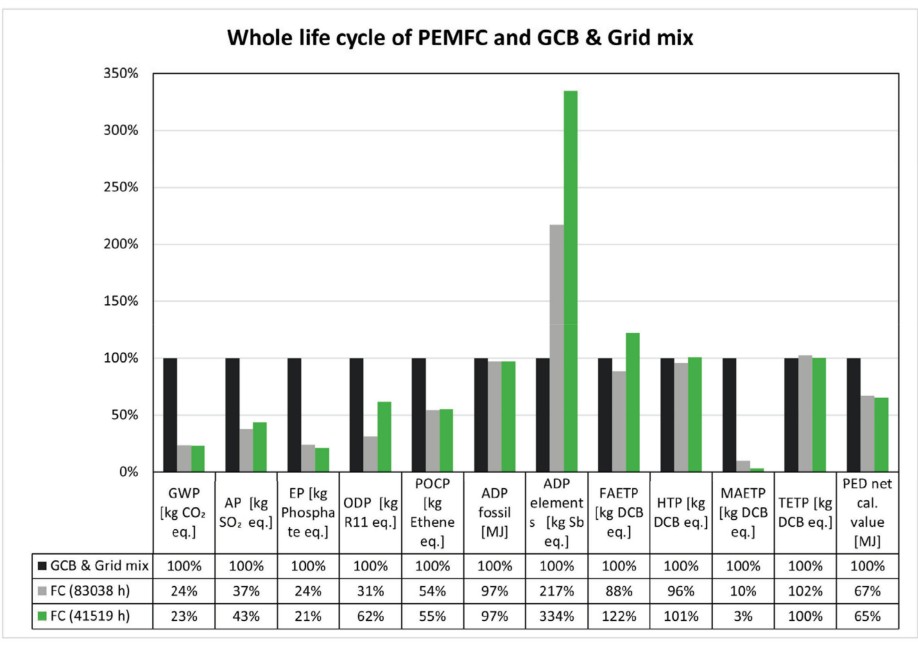

**Figure 7.** Sensitivity scenario LCA of the PEM fuel cell energy supply system: comparison of basis scenario 83,038 h (durability of the PEMFC 10 years) and the 41,519 h (durability of the PEMFC 5 years). The GCB and electricity from the grid mix system are presented as a reference system and are set as 100%.

Figure 6 presents the sensitivity analysis of the entire life cycle of the PEMFC energy supply system depending on the building type (1991–2000 as a basis scenario, old building 1949–1978 and new building from 2009). It can be observed that the environmental burdens in the old building are higher for all considered impact categories, especially for GWP (206%), ADP fossil (60%), MAETP (95%), TETP (42%), and PED (64%). In contrast, in new buildings, they are lower for GWP (41%), ADP fossil (45%), TETP (31%), and PED (32%). For the impact categories AP, EP, ODP, and MAETP, the environmental burdens in the new buildings are even higher than in the old buildings (22%, 59%, 84%, and 172%, respectively). This is explained by the fact that, in this scenario, the partial load operation (15 h per day) of the PEM fuel cell is assumed. This is due to the lower heating demand of the new energy-efficient buildings. However, the electricity produced in the partial load operation was not enough to cover the entire electricity demand of the single-family house and, therefore, should be additionally covered by the grid mix. In the old building scenario, the surplus electricity production from PEMFC was credited.

Figure 7 presents the sensitivity analysis of the entire life cycle of the PEMFC energy supply system depending on the durability (10 years—83,038 operating hours as a basic scenario and 5 years or 41,519 operating hours as an alternative scenario). The two scenarios for PEMFC durability are compared with the conventional energy supply system (GCB and grid mix), which is set to 100% as a reference system. It can be observed that the environmental burdens in the scenarios with 41,519 operating hours (5 years lifetime of the PEMFC) and 83,038 operating hours (10 years lifetime of the PEMFC) are nearly the same for GWP, AP, EP, POCP, APD fossil, HTP, TETP and PED impact categories, but significantly different for the ODP, ADP elements, FAETP, and MAETP impact categories. The nearly the same burdens of most of the considered impact categories are explained by the credits from material recycling in the end-of-life phase on the on-hand and the higher electricity credit during the use phase in the scenario with 41,519 operating hours (12,200 kWh vs. 8280 kWh) on the other hand. The contributions from electricity credits for the OPD and ADP elements during the use phase are minor. Furthermore, the impacts from ODP, ADP elements, and FAETP are higher for scenario FC (41,519 h) than for scenario FC (83,038 h). This leads back to the water tank made of low-density polyethylene. On the contrary, the MAETP for scenario FC (41,519 h) is significantly lower (3% of the contribution from conventional system GCB and Grid mix) as compared with scenario FC (83,038 h) (10% of the contribution from conventional system GCB and Grid mix) due to the credited surplus electricity for the FC (41,519 h) scenario that is higher (12,200 kWh) then for the FC (83,038 h) scenario (8280 kWh).

## 4. Discussion

In the case of the PEMFC system, a decisive tendency can be observed that the main contribution to the entire environmental profile comes from the manufacturing process except for the impact categories GWP, ADP fossil, TETP, and PED, for which the main contributor is the use phase. Focusing on the use phase, the PEMFC energy supply system shows significant environmental advantages in all impact categories except for the ADP fossil, the TETP, and PED categories as compared to conventional systems (GCB and grid mix). This is favored by the electricity credits for excited electricity produced by the PEMFC (except for the ADP fossil impact category). Considering the whole life cycle, the PEMFC energy supply system appeared to be more advantageous against the conventional system (GCB and grid mix) in almost all impact categories, except for the category ADP elements (by 117% worse) and slightly TETP (by 2% worse). This is due to the gold and platinum used in the stack and balance of the plant. It should be noted that, in this model, the primary route of gold and platinum manufacturing was assumed. Furthermore, it is to be mentioned that the methodologies for the toxicity impact categories are still not fully developed [52,53]. Therefore, the results for these categories should be treated with caution and can only be used to make a relative comparison between the two investigated systems.

The sensitivity analyses on the use phase in the different building types showed that the operation of the PEMFC system in the new buildings (after 2009) would be more environmentally friendly than in the old buildings (1949–1987) (between 11% and 206% for old building scenario) because of the much higher gas demand. Concerning the new building scenario (after 2009), advantages were found to be for 8 out of 12 considered impact categories, including GWP (41%). This result reinforces the importance of thermally well-insulated buildings with efficient energy supply systems [54]. The sensitivity analyses on the use phase under consideration of durability and degradation rate of the PEMFC system showed nearly no difference in the environmental impact for most of the considered impact categories, except for the ODP, ADP elements, FAETP, and MAETP. This is due to the counterweighing of the materials credits in the end-of-life on the one hand and the electricity credits in the use phase on the other hand. It should be noted that for this assessment, the degradation rate of ca. 1% per year was assumed (1 $\mu$V/h), which is commonly accepted for most applications. According to Arsalis et al., 2011, the operating temperature in the fuel cell stack is an important factor in the efficiency and the degradation rate of the membrane. High operating temperatures reduce the degradation rate [16]. According to the different durability testing presented in the literature, the degradation rate may vary from 1 to 20 $\mu$V/h [26], which accordingly affects the durability of the fuel cell, its environmental profile, as well as its economic feasibility and practicality.

## 5. Conclusions

To become more independent of natural gas imports and more energy self-sufficient, other green technologies, such as fuel cells, are becoming increasingly important in the EU, in addition to renewable energy systems, which can be fired highly efficiently with green hydrogen [19]. The fuel cells were widely investigated and showed proven environmental benefits compared to conventional energy supply systems for households with a gas condensing boiler (GCB). However, there are still certain environmental aspects that should be addressed, especially in terms of fuel sourcing and the use of a noble metal such as platinum or other precious metals [19]. The novelty of this study was to investigate a semi-self-sufficient system for a single-family house in Germany. That means that the household utilizes the electricity produced by the PEMFC itself and does not rely on the electricity from the grid. Hereby, we did not credit the entire amount of produced electricity, only the surpluses that are not utilized for personal needs and are therefore fed into the grid. In this case, the PEMFC results in 0.315 kg $CO_2$/kWh electricity compared to 0.506 kg $CO_2$/kWh from the grid mix. Furthermore, we analyzed different use scenarios, such as the application in different building types (old and new building types as compared to basis scenarios), as well as the scenarios for PEMFC durability. Our LCA results show that the PEMFC system can show environmental advantages compared to conventional systems, especially in the impact categories GWP and PED, whereas the PEMFC technology needs to be optimized and improved regarding other impact categories (i.e., ADP fossil, ADP elements, HTP, TETP, and FAETP). A reduction of critical raw materials (CRM) like precious metals such as platinum for the catalyst would improve the environmental performance of PEMFC, which could be further improved through material recycling and the use of secondary materials as well as eco-friendly substitutes by CRM-free catalysts (e.g., Fe-N-C-catalysts [13,19,55]).

Furthermore, the overall environmental performance of using natural gas during fuel cell operation can be improved by biogas or green hydrogen use. Hereby, it should be noted that hydrogen does not exist in large quantities in elemental form in nature. However, renewable energy from wind and solar are intermittent and are often distant from end-use appliances. To address these issues, hydrogen is increasingly being recognized as a promising renewable energy carrier, which, combined with fuel cell technology, can achieve fluctuation smoothing and peak regulation of renewable energy [56,57]. Other LCA studies showed that the use of renewable energy sources such as wind, hydropower, and solar energy, as well as water electrolysis, are more environmentally friendly methods for hydrogen production [58,59]. Moreover, it would be interesting to analyze the use of

green hydrogen from biomass sources such as biochar [10,60]. Apart from that, the change in the system scale would be an interesting research question for further studies. Such alternatives should be investigated more closely in future studies, where also substitution effects through more recycling- or eco-friendly materials should also be considered.

Furthermore, we only considered the energy supply system of a single-family house, but not the wider energy system. Therefore, further research and assessments are required to make a comprehensive comparison of the whole energy system (for example, possible expansion with photovoltaic modules, electrolyzer, and hydrogen storage tank [15]). With regard to the economic feasibility and practicability of the fuel cell, it should be mentioned that the decisive obstacles, such as reducing degradation, shortening the start-up time, and increasing power densities, must be overcome to facilitate commercialization [25,61,62]. The same applies to hydrogen production; despite the growing interest in this technology in recent years, it faces quite a few obstacles that should be overwhelming [10]. While hydrogen from conventional technologies like steam-methane reforming or water electrolysis are still high energy and cost-intensive, biomass systems are troubled by low hydrogen yield and poor stability [10].

**Supplementary Materials:** The following supporting information can be downloaded at: https://www.mdpi.com/article/10.3390/su16062348/s1; Supplementary Information (A)—Methods and analysis settings: Figure S1: (A) Investigated innovative energy supply system consisting of the proton-exchange-membrane fuel cell (PEMFC), gas condensing boiler, water reservoir, under floor heating (alternatively radiators), inverter (full load operation of the PEMFC in the basis and old building scenario and part load operation (15 h/day) in the new building scenario); (B) Investigated conventional energy supply system consisting of stand-alone gas condensing boiler, water reservoir, under floor heating (alternatively radiators). Supplementary Information (B)—Life cycle inventory of the PEMFC system: Table S1: Life cycle inventory of the PEMFC system; Table S2: Life cycle inventory of the SMA Inverter, which was manually dismantled and analyzed for the material composition; Table S3: Life cycle inventory of the checkbox Fronius, which was manually dismantled and analyzed for the material composition; Table S4: Life cycle inventory of the smart meter, which was manually dismantled and analyzed for the material composition; Table S5: Technical performance of the fuel cell heating unit Vitovalor PT 2; Table S6: Assumptions for the durability and degradation of the PEMFC based on the reviewed literature. Supplementary Information (C)—Parameterized GaBi-model of the manufacturing phase of the energy supply system: Figure S2: Flow chart of the parameterized GaBi-model of the manufacturing phase of the entire energy supply system; Figure S3: Flow chart of the parameterized GaBi-model: manufacturing phase of the PEMFC; Figure S4: Flow chart of the GaBi-model: manufacturing phase of the stack module of the PEMFC; Figure S5: Flow chart of the GaBi-model: manufacturing phase of the Balance-of-Plant module of the PEMFC; Figure S6: Flow chart of the GaBi-model: manufacturing phase of the lead battery pack module of the PEMFC; Figure S7: Flow chart of the GaBi-model: manufacturing phase of the SMA Inverter; Supplementary Information (D)—Parameterized GaBi-model of the use phase of the energy supply system: Figure S8: Flow chart of the GaBi-model: use phase of the PEMFC system (basis scenario); Table S7: GaBi Parameters for the use phase (basis scenario). Supplementary Information (E)—Parameterized GaBi-model of the End-of-Life phase of the energy supply system: Figure S9: Flow chart of the GaBi-model—EoL phase of the entire energy supply system; Figure S10: Flow chart of the GaBi-model—EoL phase: disassembling of the PEMFC; Figure S11: Flow chart of the GaBi-model—EoL phase: disassembling of the Stack; Figure S12: Flow chart of the GaBi-model—EoL phase: disassembling of the Lead Battery; Figure S13: Flow chart of the GaBi-model—EoL phase: water reservoir; Figure S14: Flow chart of the GaBi-model—EoL phase: SMA Inverter. Supplementary Information (F)—Key parameters for the description of the energy supply system of a single-family house in Germany: Table S8: Description of energy supply system of a single-family house (basis scenario). Supplementary Information (G)—Assumptions for the use phase: basis scenario and sensitivity scenarios (old and new building types): Table S9: Sensitivity analysis (building type): key figures for the use phase basis scenario and sensitivity scenarios (old and new building types); Table S10: Sensitivity analysis (PEMFC durability): Key figures for the life cycle model for the PEMFC-based energy supply system for a single-family house: basis scenario (83,038 operating hours) and sensitivity scenario (41,519 operating hours). Supplementary Information (H)—Life cycle assessment results—basis scenario: Table S11: Absolute

values for the results from comparative LCA of the energy supply system for a single-family house. Gas condensing boiler and electricity from grid mix was used as reference scenario and compared to the alternative system with a proton exchange membrane fuel cell (PEMFC); Table S12: Absolute values for the LCA results for the manufacturing phase of the PEMFC energy supply system for a single-family house in Germany; Table S13: Absolute values for the comparative life cycle assessment results for the use phase of the energy supply system for a single-family house in the German climatic conditions: gas condensing boiler and electricity from grid mix vs. the PEM fuel cell. Supplementary Information (I)—Life cycle assessment results—sensitivity scenarios: Table S14: Absolute values for the sensitivity scenario LCA of the PEM fuel cell energy supply system: comparison of basis scenario building type from 1991–2000 with the old building type from 1949–1978 and new building type from 2009; Table S15: Absolute values for the sensitivity scenario LCA of the PEM fuel cell energy supply system: comparison of basis scenario 83,038 h (durability of the PEMFC 10 years) and the 41,519 h (durability of the PEMFC 5 years). The GCB and electricity from grid mix system is presented as a reference system and is set as 100%.

**Author Contributions:** Conceptualization, L.S., F.A., F.P. and M.-C.S.; L.S. conducted the literature research and review, data curation, performed model development, calculations, and analysis, writing—original draft preparation. Writing—review and editing, L.S., F.A., F.P. and M.-C.S.; supervision, project administration, F.A. All authors have read and agreed to the published version of the manuscript.

**Funding:** This research was funded by the German Energy and Climate Fund as part of the German National Action Plan for Energy Efficiency (NAPE) and carried out at the Federal Institute for Materials Research and Testing (BAM).

**Institutional Review Board Statement:** Not applicable.

**Informed Consent Statement:** Informed consent was obtained from all subjects involved in the study.

**Data Availability Statement:** Data is contained within the article and Supplementary Materials [63–65]. The data that support the findings of this study are available from Sphera. Restrictions apply to the availability of these data, which were used under license for this study. Data are available from the Sphera at Sphera | Sustainability, Operational Risk Management and EHS Software: Sphera LCA for experts (former GaBi—Product Sustainability Software (version 10.6.2.9)) with the permission of Sphera.

**Conflicts of Interest:** The authors declare no conflicts of interest.

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
