# Peer review of "Life Cycle Assessment of the Domestic Micro Heat and Power Generation Proton Exchange Membrane Fuel Cell in Comparison with the Gas Condensing Boiler Plus Electricity from the Grid"

_sustainability, doi:10.3390/su16062348_

Round 1

Reviewer 1 Report

Comments and Suggestions for Authors

Attached please find the comments.

Reviewer 2 Report

Comments and Suggestions for Authors

This manuscript reported and conducted a comparative life cycle assessment of the PEMFC in a residential application with a conventional system with the stand-alone gas condensing boiler and electricity from a grid mix to analyze the emissions reduction potential. The impact assessment method “CML 2001 - Aug. 2016” was adopted. The manufacturing phase of the PEMFC showed disadvantaged, whereby the use phase – significant advantages in the most of environmental impact categories as compared to the conventional energy supply system. Considering the whole life cycle, it is found that the advantages from the use phase could outperform the disadvantages from the manufacturing phase in most of the impact categories.

The content of the manuscript is within the scope of the journal and can be of broad interest to readers. However, in terms of specific content, there is still room for improvement. Therefore, I decided to give the decision of minor revision. It is recommended that the author properly absorb the reviewers' comments and make corresponding improvements and enhancements.

1. For the keywords, 'micro heat and power generation' and 'residential application' should be added to attract a broader readership. Moreover, 'conventional energy supply system 5, innovative energy supply system 6' should be merged, since they are quite similar.

2. Page 2, 'The comparative LCA results showed that the environmental impacts increase in the case of both fuel-cell systems (alkaline water electrolyser (AWE) and proton-exchange-membrane water electrolyser (PEMWE)) because of the larger quantity of PGMs compared to the scenario without the recycling of PGMs [17].'

    These are not true. The authors wrote fuel cell systems, but indeed AWE and PEMWEis water electrolyser. This is to produce hydrogen, not fuel cell systems to use hydrogen and convert chemical energy to electricity.

3. Page 2, 'Parise et al. concludes that there is still long way to go for the widespread use of fuel cells, which depends on cost, efficiency, and life expectancy [16].' 

    Moreover, the mass transfer in the PEMFC system should be well controlled since they influence the final performance significantly. For example, the large-scale commercialization of PEMFCs requires higher power and current densities; however, at high operating current densities, the massive accumulation of liquid water will lead to flooding and impede the gas diffusion, resulting in rapid degradation of cell performance. Accordingly, improving the water management ability is imperative for pursuing better cell output performance (10.3390/en16166010).

4. Page 4, 'The fuel cell was proton-exchange-membrane fuel cell high temperature (PEMFC-HT) system with the lead battery as an intermediate storage, 5 kW and an assumption of linear scaling with a scaling factor 0.15 to 750 W electrical power was undertaken [17].'

    The HT-PEMFC adopts PBI membrane rather than perfluoro-sulfonated membranes such as Nafion for LT-PEMFC, and the commercialization of HT-PEMFC is currently not as good as LT-PEMFC. Why is this system selected in this study? Some explanations should be provided. And for the battery, why is lead battery selected not a lithium-ion one? This issue should also be further clarified.

5. Page 11, 'Figure 7 presents the sensitivity analysis of the entire life cycle of the PEMFC energy supply system depending on the durability (10 years – 83038 operating hours as a basis scenario, and 5 years or 41519 operating hours as an alternative scenario).'

    The durability of HT-PEMFC depends on which parameters or factors should be explained briefly, which may give guidelines for the usage of HT-PEMFC to avoid excess degradation and overuse of the HT-PEMFC.

6. Page 12, 'This is lead back to the use of higher amount of the low-density polyethylene in the balance of plant components of the PEMFC'. 

    It should be explained why higher amount of the low-density polyethylene is used, and where these LDPE is used in the fuel cell. They are used for structured material, or for the membrane material, or for other components should be explained and confirmed to avoid confusion.

7. Page 13, 'Furthermore, the overall environmental performance using natural gas during the fuel cell operation can be improved by biogas or green hydrogen use because other LCA studies showed that the use of renewable energy sources such as wind, hydropower, and solar energy, as well as water electrolysis are more environmentally friendly methods for hydrogen production [36, 37]'.

    I think it should be emphasized here that hydrogen does not exist in large quantities in elemental form in nature. However, most renewable energy sources such as wind and solar have intermittent properties, opening spatial and temporal gaps between the availability of the energy and its consumption by end users (10.1016/j.electacta.2019.03.056). To address these issues, it is necessary to develop suitable energy conversion and storage systems for the power grid. Therefore, using renewable energy to electrolyze water and produce green hydrogen combined with fuel cell technology for power generation can achieve fluctuation smoothing and peak regulation of renewable energy.

Reviewer 3 Report

Comments and Suggestions for Authors

It would be advised that the authors of the article clearly and in a logically consequential manner state the design of the research, hypotheses, and research questions. Without them clearly stated, it is rather problematic to evaluate the logical coherence of the research. It would also help the reader navigate the text much easier. It is confusing that the authors write, “PEMFC is assumed to be 10 years, it was operated in a single-family house from 1991 to 2000 in German climatic conditions." The choice of operational period for the installation could seem a bit out of place, and it would be advisable to change it to a more recent one. If the average annual heat and electric energy consumption (obviously, higher than in the last 10–15 years) is a key factor in choosing the period, it should be stated simply that the analysis is made in accordance with average annual energy consumption data from 1991 to 2000.

The authors should consider adding more information about the lifecycle approach, which is mentioned throughout the research but is not detailed and summed up in one single chapter, which would be advisable.

Reviewer 4 Report

Comments and Suggestions for Authors

Thank you for your contribution. I was able to understand your underlying model. I was of course very impressed by the entire appendix (Supplementary document), which made the modelling comprehensible. You also describe that the results will of course change due to new technologies and with correspondingly changed values. However, I would ask you to consider whether you could summarise the supplementary document somewhat.

Reviewer 5 Report

Comments and Suggestions for Authors

The minor polishing of the manuscript should be done before it can be accepted for publication.
1. It should be a lower register in chemical compound formula CO2 not CO2.
2. Fuel cell abbreviation should be added in the 52 line.
3. The abbreviation PEMFC is in the 85 line while its explanation is in the 94 line (after it's usage).
4. It is worth bringing the number of the figure /table in the supplementary and its mention in the text into an agreed form. For example, Table S4 is mentioned before Tables S1-S3 and so on.
5. Some figures (e.g. 3 and 5) mentioned and discussion in text could be observed after these figures, but should be before.

Reviewer 6 Report

Comments and Suggestions for Authors

This research consists of a large amount of data that has been analyzed and presented through various impact categories. Despite certain benefits of fuel cells compared to conventional energy supply systems, the authors also pointed out some of their objective disadvantages.
The article represents a contribution to the research of new self-sufficient energy supply systems for a single-family house.

Round 2

Reviewer 1 Report

Comments and Suggestions for Authors

I am very satisfied with the updated and revised version of the manuscript. The authors have greatly improved the quality of the data presented and have addressed all my comments and concerns. I therefore recommend the publication of this research study in its present form.